# Nanomedicine for the Delivery of RNA in Cancer

**DOI:** 10.3390/cancers14112677

**Published:** 2022-05-28

**Authors:** Michele Ghidini, Sandra G. Silva, Jessica Evangelista, Maria Luísa C. do Vale, Ammad Ahmad Farooqi, Marina Pinheiro

**Affiliations:** 1Medical Oncology Unit, Fondazione IRCCS Ca’ Granda Ospedale Maggiore Policlinico, 20122 Milan, Italy; michele.ghidini@policlinico.mi.it; 2LAQV/REQUIMTE, Department of Chemistry and Biochemistry, Faculty of Sciences, University of Porto, 4169-007 Porto, Portugal; sandra.silva@fc.up.pt (S.G.S.); mcvale@fc.up.pt (M.L.C.d.V.); 3Thoracic Surgery, Fondazione Policlinico Universitario A. Gemelli IRCCS, Università Cattolica del Sacro Cuore, Largo F. Vito 1, 00168 Rome, Italy; evangelistajessica664@gmail.com; 4Institute of Biomedical and Genetic Engineering (IBGE), Islamabad 44000, Pakistan; farooqiammadahmad@gmail.com; 5REQUIMTE, University of Porto, 4169-007 Porto, Portugal; 6ICVS, Life and Health Sciences Research Institute, School of Medicine, University of Minho, 4710-057 Braga, Portugal

**Keywords:** cancer, drug delivery systems, RNA, nanoparticles

## Abstract

**Simple Summary:**

Cancer is a multifaceted, life-threatening, and genomically complex disease. The worldwide prevalence of cancer is so high that one in three people will develop cancer during their lifetime. Although the use of RNA therapy is promising to fight cancer, its efficient and safe delivery is still one of the significant challenges hampering its therapeutic application. Thus, the aim of the present review was to highlight the most recent developments in the field of nanomedicine RNA-associated therapies to fight cancer.

**Abstract:**

The complexity, and the diversity of the different types of cancers allied to the tendency to form metastasis make treatment efficiency so tricky and often impossible due to the advanced stage of the disease in the diagnosis. In recent years, due to tremendous scientific breakthroughs, we have witnessed exponential growth in the elucidation of mechanisms that underlie carcinogenesis and metastasis. The development of more selective therapies made it possible to improve cancer treatment. Although interdisciplinary research leads to encouraging results, scientists still have a long exploration journey. RNA technology represents a promise as a therapeutic intervention for targeted gene silencing in cancer, and there are already some RNA-based formulations in clinical trials. However, the use of RNA as a therapeutic tool presents severe limitations, mainly related to its low stability and poor cellular uptake. Thus, the use of nanomedicine employing nanoparticles to encapsulate RNA may represent a suitable platform to address the major challenges hampering its therapeutic application. In this review, we have revisited the potential of RNA and RNA-associated therapies to fight cancer, also providing, as support, a general overview of nanoplatforms for RNA delivery.

## 1. Introduction

With increasing awareness of the interdisciplinarity needed for a comprehensive characterization of the underlying mechanisms of cancer research, we have witnessed groundbreaking discoveries in various facets of molecular oncology. Compelling experimental evidence obtained from high-throughput technologies has offered a window into previously intractable problems in our comprehension of cancer genetics/epigenetics, deregulated cell signaling pathways, noncoding genome, and tumor heterogeneities and provided new insights into therapeutic options. Rapidly evolving understanding has shown that by gaining deeper insights into nano-bio interactions and personalization of nanomedicines, and through the applications of nanotechnology to emerging and existing therapeutic modalities, we have started to realize the true potential of nanomedicines in cancer [1,2,3,4].

Recent studies have provided evidence of an upsurge in cytokines after the administration of positively charged nanoparticles. There is sufficient proof of the correlation of Compliment activation with nanoparticle administration. Importantly, nanoparticles with a positive surface charge triggered activation of the classical compliment pathways, whereas negatively charged particles “switched on” the alternative (lectin) pathways [5,6].

It is becoming sequentially more understandable that after intense experimental and clinical evaluation of protein biologics and small molecules, gene therapy and RNA medicines represent promising models of drug innovation. RNA vaccines against different cancers showcase an efficient technology, as they are easier and faster to develop and manufacture compared to conventional vaccines. Importantly, RNA vaccines are completely synthetic and do not require cell cultures.

In this review, we have attempted to highlight the most recent developments in the field of nanomedicine.

## 2. RNA for Cancer Therapy

RNA therapy acts on messenger RNA (mRNA) by using oligonucleotides that can interfere with different metabolic processes of a polynucleotide, such as splicing, the mature process starting from pre-mRNA, transport, translation, and degradation [1] Differently from standard chemotherapy, RNA therapy harbors high specificity and may be used to target multiple critical oncogenic drivers, reduce drug resistance of tumor cells, and arrest growth of advanced-stage tumors [2]. RNA therapeutics may act through several mechanisms. They can inhibit the proliferation and induce apoptosis of tumor cells, prevent the metastasization process, disrupt the gene’s expression, inhibit angiogenesis, reconstruct the tumor environment, reprogram, and decrease drug resistance of tumor cells [2] RNA therapy may be divided into three major classes: antisense oligonucleotides (ASO), RNA interference (RNAi) therapies, and messenger (mRNA) therapy (Figure 1 and Table 1). ASO are single-stranded sequences of 15–25 nucleotides that bind specifically to target mRNA by complementary base pairing. Because of ASO’s weak hydrophilicity, a common modification is thiolization to increase diffusion in tissues and absorption.

Their easy diffusion and absorption through tissues allows them to directly bind to targets after being injected into patients [7]. 

ASO may act at their target mRNAs by activating RNase H or inhibiting translation by preventing ribosomes’ action through a steric effect [2]. ASO may act by correcting an altered spliceosome of proteins, repairing defective RNAs, restoring proteins, or downregulating genes’ expression [2]. Among ASO, some act as miRNA inhibitors. These oligonucleotides bind to the active chains of endogenous miRNAs with gene-silencing effects. Therefore, they enhance gene expression [2]. In contrast, RNA interference (RNAi) therapy is triggered by double-stranded RNA (dsRNA). RNAi therapy acts by knocking down the expression of the genes of interest by promoting short interfering RNAs (siRNAs). Some molecules are of synthetic manufacturing (siRNA and specific RNAi sequences). Moreover, RNAi therapy may be delivered through short hairpin RNAs (shRNAs) and microRNAs (miRNAs) [2]. siRNAs are double-stranded RNA molecules, 20–25 nucleotides in length. They are made from cutting a long dsRNA and disrupting mRNA before translation by binding it with 100% complementarity and high target specificity. Short hairpin RNAs (shRNAs) are sequences of RNA, typically about 80 base pairs in length, that include a region of internal hybridization that creates a loop structure. shRNA molecules are processed within the cell upon transcription to form a double-stranded siRNA, which knocks down gene expression.

**Table 1 cancers-14-02677-t001:** Main classes of RNA therapy.

Class of RNA Therapy	Features	Example in Cancer Therapy (Formulation)	Target	Indication	References
ASO	12–25 nucleotides Single-stranded Chemically modified (3 classes)	Danvatirsen	STAT3 (downregulation)	Advanced/recurrent solid tumors or lymphoma	[8]
siRNAs	20–25 nucleotides Double-stranded Incorporated in RISC	ALN-VSP02 (lipid nanoparticle-formulated)	VEGF and KSP (downregulation)	Solid tumors with liver involvement	[9]
miRNAs	18–25 nucleotides Single-stranded Incorporated in RISC	miR-29b (cationic lipoplexes)	CDK6, DNMT3B, MCL1 (downregulation)	Lung cancer	[10]
miRNA mimics	18–25 nucleotides Single-stranded Incorporated in RISC	miR-4689	*KRAS*, *AKT* (downregulation)	KRAS mutant colorectal cancer	[11]
anti-miR	18–25 nucleotides Single-stranded Incorporated in RISC	Anti-miR-155	miR-155 (downregulation)	Colorectal cancer	[12]
shRNA	80 nucleotides Double-stranded with a loop sequence Incorporated in plasmid vectors	hTERT-shRNA (plasmid)	hTERT (downregulation)	Colorectal cancer	[13]
mRNA	Single-stranded Less stable than DNA	AGS-003 (dendritic cells)	CD40L RNA tumor RNA	Renal cancer	[14]

Legend: RNAs; shRNA: short hairpin RNA; siRNA: small interfering RNA; RISC: RNA-inducing silencing complex.

The benefit of shRNA is that it can be incorporated into specific plasmid vectors, permitting cell-type-specific or inducible promoters integrated into genomic DNA for longer-term or stable expression, and thus more prolonged knockdown of the target mRNA [3]. miRNAs are short-endogenous non-coding RNA molecules that limit gene expression by restricting mRNA from translation and promoting mRNA decay. MiRNAs are integrated with siRNAs and proteins in the RNA-induced silencing complex (RISC). They regulate gene expression based on pairing with the target mRNA’s 3′ untranslated region (UTR). When binding with complementary mRNA occurs, RISC activates its RNase component and degrades its target. On the other hand, miRNAs come from single-stranded RNA. Folding miRNAs create stem-loops, small, folded areas of dsRNA [4]. Due to imperfect base pairing, miRNA action can affect hundreds of less specific genes. Moreover, miRNA may influence the CpG island methylation of gene promoters and regulate gene expression at the transcriptional level [2]. Among RNAi techniques, miRNA mimics function similarly to endogenous miRNAs. Their action may restore altered miRNA maturation mechanisms or enhance the function of specific tumor suppressor miRNAs [5]. In contrast, miRNA competitive agonists block the binding of endogenous miRNAs to RISC. In this way, they upregulate the expression of related proteins [6]. mRNA therapy is an alternative to DNA therapy and consists of the injection of a specific RNA messenger into a patient’s body to promote protein synthesis in cells [15]. mRNA is less stable than DNA and is an easy target of endogenous nucleases, with possible immune responses given by high numbers of neoantigens. To reduce these events, modification of the nucleoside portion of the uracil ribose has been evaluated with the creation of an immune-evasive “pseudouracil” [16]. Both siRNAs and miRNAs have hydrophilic nature, negative charge, and relatively high molecular weight (14–15 kDa), which make them poorly permeable across biological membranes. However, encapsulation of siRNAs into vesicles or conjugation to certain ligands can help deliver them to desired tissues or cells and, at the same time, avoid renal clearance [17]. With respect to siRNAs and miRNAs, mRNAs have higher and heterogeneous molecular weights and are negatively charged. Unlike drugs that can cross the lipid bilayer, the vast majority of RNA-based therapeutics are too charged and/or too large to enter cells, and demand a delivery agent. Nanoparticles (such as liposomes, polymers, and peptides) have been used to shuttle mRNA to the cell cytosol [18]. In addition, the use of lipid nanoparticles (LNPs) as a delivery system for mRNA allows the extended time of drug action, reduced drug toxicity, and improved drug stability [19]. Table 1 shows the different classes of RNA therapies and gives examples for each of them.

## 3. Nanoparticles for the Delivery of RNA

The development of RNA-based therapeutics has experienced a boost since the 1990s owing to the increasing knowledge of nucleic acid chemistry and the decline in production costs of mRNA [20].

The therapeutic potential of RNA largely depends on its ability to reach the desired target cells and express the proteins of interest. However, RNA presents limited stability in serum, suffers from rapid blood clearance, off-target effects, and poor cellular uptake, and may activate immune responses [21,22,23]. Therefore, the efficient and safe delivery of RNA is still one of the significant challenges hampering its therapeutic application.

The translation efficiency and stability of exogenous RNA can be enhanced by several methods, such as UTR (untranslated regions) manipulation, codon optimization, and chemical modification of the poly(A)tail of RNA [24,25]. Furthermore, its immunogenicity can be reduced through high-performance liquid chromatography purification and chemical manipulation [26,27,28,29,30,31,32,33]. Nevertheless, the optimized RNA still has to avoid enzymatic degradation, interact with the target cell, cross the cytoplasmic membrane, and diffuse in the cytoplasm to reach the ribosomes. Despite these modifications, RNA therapeutics fail to show efficient and specific uptake by tumor cells. Recent advances in nanotechnology have led to new opportunities in cancer prevention and treatment. Novel formulations of RNA in nanosystems or vectors were developed [34]. The use of viral and non-viral delivery systems resulted in improved stability and toxicity, tumor-specific delivery, and reduced immunogenicity. Viral delivery systems (retroviruses, lentiviruses, adenoviruses) make up about two-thirds of clinical trials with nucleic acids performed to date. Although effective in targeted cellular delivery, these systems raise some safety concerns related to immune responses, have a reduced cargo capacity, and are difficult to scale up manufacturing [35,36,37,38]. Non-viral delivery systems have emerged as a safer alternative for cancer therapy, as they are less immunogenic, less toxic, and less oncogenic. In addition, their production is more cost-effective and easier to scale up [39]. However, these systems’ low nucleic acid delivery efficiency hampers their translation into clinical practice and represents a problem that still needs to be addressed [40]. Nowadays, there exists a vast array of non-viral nanocarriers for RNA delivery, with distinctive compositions and thus unique properties. Some essential nanoparticle platforms are liposomes, exosomes, polymers, dendrimers, nanogels, and inorganic nanoparticles, such as carbon nanotubes and gold and magnetic nanoparticles (Figure 2) [41].

### 3.1. Lipids or Lipid-Based Nanoparticles

One of the most advanced RNA delivery methods is co-formulation into lipid nanoparticles (LNP) [42,43]. LNP for RNA delivery mainly comprises a cationic or ionizable lipid bearing a tertiary or quaternary ammonium group, which encapsulates the polyanionic RNA, protecting it from degradation and increasing its stability in blood circulation (Figure 3). In addition to the cationic/ionizable lipid, these formulations typically contain a zwitterionic lipid (helper lipid, e.g., 1,2-dioleoyl-sn-glycero-3-phosphoethanolamine, DOPE) mimicking cell membrane lipids, cholesterol for the stabilization of the lipid bilayer of the nanoparticle, and a polyethene glycol (PEG), meant to improve colloidal stability and reduce protein absorption [44,45]. Cationic lipids frequently used in lipoplexes include N-[1-(2,3-dioleyloxy)propyl-N,N,N-trimethylammonium chloride (DOTMA) and N-[1-(2,3-dioleoyloxy)propyl]-N,N,N-trimethyl-ammonium chloride (DOTAP). The use of cationic lipids for lipoplex formation enhances the uptake of RNA through the interaction of the positively charged complexes with the negatively charged cell membranes. Several cationic LNP have been successfully used as RNA carriers in targeted cancer therapy, leading to higher accumulation and increased protein expression, which resulted in suppressed/blocked tumor growth [10,46].

The inclusion of helper lipids, such as DOPE or cholesterol, generally increases the in vitro transfection efficiency of the lipoplexes [47]. However, most of these positively charged systems are highly cytotoxic in vivo, and the formulations have to be carefully adjusted to maintain cellular viability [48,49,50,51].

More recent studies have focused on using LNP based on pH-dependent ionizable cationic lipids, which have been shown to efficiently transfect mRNA to express therapeutic proteins. These lipids are positively charged at acidic pH but neutral at physiological pH. The resulting LNP display low surface charge at physiological pH and are relatively non-toxic and non-immunogenic. The structure of pH-dependent nanoparticles may be destabilized in environments with pH values lower than 6.5, facilitating the release of encapsulated cargoes within the more acidic tumor microenvironment [52,53,54]. In fact, the first siRNA drug approved by the FDA, Onpattro, is based on ionizable lipid (6*Z*,9*Z*,28*Z*,31*Z*)-heptatriaconta-6,9,28,31-tetraen-19-yl-4-(dimethylamino) butanoate (DLin-MC3-DMA, MC3) [55]. Several MC3-based LNP have then been tested for mRNA therapeutics [56,57]. Ethanolamine was identified as a favorable headgroup. The incorporation of biodegradable lipids resulted in nanoparticles with reduced toxicity and better delivery efficacy [58,59]. Biodegradability may be conferred by the presence of an ester bond on the hydrophobic tail or on the linker, which accelerates liver clearance. In addition, the release of the cargo may be triggered through cleavage of the labile ester bond (pH, nucleases) and consequent modification of the aggregate structure [58,59,60].

Further, the use of unsaturated lipid tails in the lipid structure has been shown to increase fluidity and introduce structural defects in the cell membrane, facilitating fusion of the LNP with the cell membrane, as well as endosomal escape [61]. However, a rational balance of unsaturation and biodegradability is of the utmost importance, since these two factors seem to strongly affect the degree and site of protein expression [62,63].

The development of lipid nanoparticles based on serine-derived gemini surfactants and monoolein (MO) as helper lipids for siRNA delivery has also been reported. The use of amino acids as polar headgroups in the design of surfactants leads to enhanced biological properties (biocompatibility and toxicity) compared to conventional quaternary ammonium-based surfactants [64]. The lipoplexes formed (gemini/MO/siRNA) were found to have sizes of 100–250 nm and were suitable for intravenous administration. The systems were effective in RNA complexation and gene silencing and presented no significant cytotoxicity. The transfection efficiency was shown to be dependent on the content of the MO.

Solid lipid nanoparticles (SLN) containing cationic lipids were also reported as RNA carriers for cancer therapy. They can be produced without the need for organic solvents, lyophilized, and the dehydrated SLN are stable for up to 9 months when stored at temperatures up to 30 °C [65]. In addition, the lyophilized SLN maintained their transfection efficacy over time [66]. Although effective as delivery systems, several problems associated with the positive charge of the lipids (e.g., toxicity) precluded their development and clinical use [67]. When neutral lipids were used to replace cationic ones, significant RNA accumulation, target genes’ downregulation, and tumor growth inhibition were achieved without inducing toxicity. However, these neutral lipid SLN suffer from lower loading capacity and lower transfection efficiency compared to the cationic lipid SLN.

More recently, miRNA-loaded exosomes have been engineered as cancer therapeutics (endometrial cancer, breast cancer, colorectal cancer, liver cancer) [68,69,70,71,72]. Exosomes are small (50–150 nm) endogenous membrane vesicles secreted from several mammalian cell types. These extracellular vesicles (EV) can fuse with the membrane of target cells and deliver exosome surface proteins, carbohydrates, lipids, and nucleic acids. They are non-immunogenic and non-oncogenic, present negligible toxicity, and thus stand as a promising and innovative platform for miRNA delivery [73]. Shtam et al. have shown that exosomes can efficiently deliver siRNA into target cancer cells, leading to gene silencing and cancer cell death [74]. However, there are still several limitations to implementing an EV-mediated miRNA cancer therapy, mainly related to the large-scale production, isolation, and characterization of EV suitable for clinical translation studies. In addition, the determination of the RNA content in the EV faces some problems. Finally, the dosage must be accurately defined, and routes of administration must be better explored since most systemically injected EV are delivered to the liver [75].

### 3.2. Polymers

Polymeric-based non-viral vectors represent another class of nanoscale platforms for RNA delivery. Specifically, cationic polymers can bind to nucleic acids to form polyplexes. Polymers can efficiently protect RNA from nucleases and promote cellular uptake and endosomal escape, leading to higher RNA delivery efficiency. Representative polymers of this class include chitosans, polyethyleneimine (PEI), dendrimers, and nanogels (Figure 2).

*Chitosans* are naturally derived cationic polysaccharides, differing in the degree of *N*-acetylation and molecular weight (50–2000 kDa). They are readily available, biodegradable, easy to modify, and possess unique biological properties associated with their polycationic nature [76,77,78]. Chitosan is only poorly water-soluble and exhibits low transfection efficacy. However, it can be derivatized to increase nucleic acid delivery efficiency by chitosan vectors. Strategies for derivatization include structural modifications–like (i) copolymerization: polyethylene glycol, PEG, and polyethyleneimine, PEI, are commonly used, although other chitosan graft copolymers are being studied as nucleic acid carriers [79]; and (ii) functional group modification: *N*-alkylation and quaternization enhance colloidal stability and transfection efficacy of the nanoparticles–and ligand conjugation–peptides, proteins, and non-proteinaceous ligands, like carbohydrates, folic acid and hyaluronic acid are commonly used for chitosan vector conjugation [80,81]. Although non-proteinaceous ligands are usually less immunogenic and produce more stable vectors, proteinaceous ligands offer a vast diversity of choices with favorable functionalities to vector conjugation for nucleic acid delivery [82]. Chitosan-based nanoparticles simultaneously encapsulating siRNA and the anticancer drug doxorubicin were shown to decrease the viability, growth, proliferation, and migration of breast [83] and colorectal [84,85] cancer cells and induce their apoptosis. 

*PEI* is a cationic polymer that has been widely used in gene therapy [86]. It is highly efficient in the compaction of nucleic acids and promotes endosomal escape via the proton sponge effect [87]. The transfection efficiency of PEI-mRNA polyplexes in several types of cells has been reported, but cytotoxicity from PEI hinders its therapeutic application [44,48,88].

Incorporating PEI into polymers (as above for chitosan) has been reported to decrease its toxicity. For example, PEI-β-cyclodextrin conjugates show low toxicity while being highly efficient in promoting cell uptake and endosomal escape, leading to enhanced transfection efficiency [89].

Chemical modification of low molecular weight PEI (<2 kDa) with salicylamide or stearic acid has also been reported to reduce the toxicity and enhance transfection efficiency of the resulting polyplexes/micelles [90,91].

*Dendrimers* are polymeric materials with a highly branched 3D structure [92]. They consist of a central core, many layers of repeating units, and multiple functional groups on the surface. Due to their unique structure, they possess various interesting physical properties, such as good water-solubility, nanoscale uniform size, symmetrical shapes, internal cavities, good biocompatibility, stability, and high drug-loading capacity. Their toxicity is generally low but depends on the number of terminal amino groups and positive charge density. Dendrimer-based organic and inorganic nanoparticles have been widely studied and exhibited high potential in cell targeting and drug delivery [93,94,95]. Dendrimer-based iron oxide nanoparticles (IONP) show reduced toxicity, enhanced biocompatibility and safety, enhanced escape from the reticuloendothelial system (RES), and enhanced MRI properties compared to non-functionalized IONP. Dendronized magnetic IONP have been applied in gene delivery as magnetoplexes for magnetofection, leading to high-level transgene expression after a short incubation time using low doses of nucleic acid and a small amount of NP [96,97,98].

Dendrimers have also been used as templates for gold nanoparticles (AuNPs) to control their size and shape, which are known to influence the efficiency of AuNPs in biological systems highly [99,100].

*Nanogels* (NG) are gel particles with a three-dimensional hydrophilic network structure. They are commonly made up of polyacrylic acid, polyacrylamides, polyaminoacids, and other high molecular weight polymers. NG-based drug delivery systems possess a large surface area, structural stability, and the ability to swell. They present a high loading capacity, encapsulating either hydrophobic or hydrophilic drugs. Their shape and size can be finely tuned, and they are sensitive to pH, temperature, ionic strength, and other external stimuli, which confers them adequate controlled drug release capacities [101,102,103].

A thiolated PEI-dextrin NG was shown to efficiently deliver siRNA to cancer cells without inducing compromising toxicity. In any case, the variable morphology of the nanoparticles, as well as their size distribution, allied to the lack of data on their clinical safety and efficacy, are issues to be overcome for a broad application of this nanoplatform to be feasible [104].

### 3.3. Inorganic Nanoparticles

*Inorganic nanomaterials* (derived from gold, carbon, silica, etc.) are promising carrier platforms for RNA delivery due to their unique physicochemical properties, which endow them exciting attributes, such as long-term stability, high loading capacity, and optical responsiveness. These inorganic nanoparticles are usually easier to synthesize and scale up than organic-based ones and have been the subject of much research concerning their potential as nanocarriers for nucleic acid delivery. 

The calcium phosphate (CaP) composite is the oldest non-viral gene carrier, introduced in 1973. It is biocompatible and biodegradable and forms complexes with nucleic acids, successfully delivering them to cells [105]. However, the size of the CaP precipitates is challenging to control, which constitutes a severe limitation to their use. Pegylation and lipid coating improve these nanocarriers’ colloidal stability, which shows relevant in vivo efficacies [106,107,108,109]. The calcium-phosphate core is responsible for endosomal escape and cargo release into the cytosol. Furthermore, these nanoparticles can efficiently co-deliver siRNA and mRNA [110].

*Gold nanoparticles*, AuNPs, possess high chemical stability and attractive optical properties, are easy to functionalize, and are of particular interest for biomedical applications [111]. They have been applied as therapeutics in drug delivery, diagnostics, and imaging [112]. Their efficiency in biological systems depends mainly on their shape, size, and size distribution, and many approaches have been made to optimize these parameters [99,100,113]. Gold can be directly conjugated via electrostatic and/or covalent interactions to thiolated compounds to form stabilized monolayer-coated NP, whose properties can easily be tuned through functionalization to meet specific needs. For example, Ghosh et al. developed efficient nanocarriers based on cysteamine-functionalized AuNP to deliver miRNA to cancer cells, using neuroblastoma and ovarian cancer cell lines [114].

*Carbon nanotubes* (CNT) can be categorized into single-, double-, or multi-walled (SWNT, DWNT, and MWNT) according to the number of graphene layers in their structure. They are promising nanocarriers for nucleic acid delivery, as they can avoid the endosomal barrier through an endocytosis-independent cell penetration pathway. CNT are poorly water-soluble; however, they offer a large surface area, which can be modified with functional groups and loaded with drugs or nucleic acids, thus enhancing their aqueous dispersibility [115,116]. CNT-mediated delivery of siRNA may be accomplished either by chemical conjugation of the nucleic acid to the CNT or the material used to coat them [117] or through formation of non-covalent complexes between chemically functionalized cationic CNT and the negatively charged siRNA [118,119,120,121]. Surface modification of CNT using dendrimers has been reported to enhance their aqueous dispersibility. These nanocarriers have been shown to efficiently complex siRNA and mediate its intracellular delivery with minimal induced cytotoxicity, and may thus be good candidate vectors for in vivo gene silencing [122].

*Mesoporous silica-based nanoparticles* (MSNP) were first applied for drug delivery in 2001 [123]. They present low toxicity, large surface area, and thus an enhanced loading capacity. However, they cannot induce endosomal escape and must be chemically modified to enhance their transfection efficacy in vivo [124,125]. Ngamcherdtrakul et al. modified siRNA loaded MSNP by adding PEI to promote endosomal escape, and PEG, to protect the siRNA from degradation, and reduce the toxicity induced by the PEI. The modified nanoparticles were shown to induce apoptosis in breast cancer cells in vitro [126]. Functionalization of MSNP with cyclodextrin-grafted PEI has also been reported to enhance the loading capacity of siRNA and enable its effective endosomal escape [127]. Furthermore, silica nanoparticles conjugated to a disialoganglioside antibody were used to deliver miRNA to neuroblastoma, with promising results [128].

*Iron oxide magnetic nanoparticles* have also been proposed as vehicles for the delivery of nucleic acids [129]. Magnetic nanoparticles coated with positively charged polymers, such as PEI, have been shown to significantly increase transfection efficiency compared to cationic polymers/surfactants or lipids alone [130]. This technique, known as magnetofection, relies on the combination of magnetic nanoparticles with a positively charged coating and nucleic acids to form the corresponding complexes, which upon applying a magnetic field adhere to the cell surface and internalize by endocytosis [131,132]. Nanoparticle coating has been reported to protect nucleic acid from degradation by nucleases. Superparamagnetic iron oxide nanoparticles (SPION) are effective vehicles for the delivery of nucleic acids and offer the possibility of monitoring biodistribution. Luo et al. reported the use of a folic acid functionalized polyethyleneimine SPION for siRNA delivery to gastric cancer cells [133].

The methods described above represent some of the most used nanoplatforms for RNA delivery. Still, others exist that were not referred to, as the aim was to provide a general, although not extensive, overview of the wide variety of available delivery systems, specifically for RNA delivery. The classifications of the delivery systems may sometimes overlap, as, e.g., in the case of inorganic gold nanoparticles functionalized with polymers. Whether they fit in the class of AuNP, or of polymers, or may be considered as hybrid nanoparticles, the most important thing is to be aware of the many considerations that have to be taken into account when designing novel vectors for biomedical applications. Efficacy concerning the proposed aim must go hand-in-hand with biocompatibility and safety; otherwise, translation into clinical use may be at cost.

In Table 2, some of the advantages and disadvantages of the above-mentioned nanosystems are summarized.

## 4. Ongoing Clinical Trials Using RNA-Loaded NPs for Cancer Therapy

Lipid-based nanoparticles (LNPs) that include liposomes and lipid NPs have been tested in animal models of tumors with promising results [9]. Therefore, clinical studies evaluating the role of RNA molecules encapsulated in LNPs have been conducted. So far, only LNPs, and no other type of nanoparticles have entered clinical trials for the delivery of RNA. Few studies have been concluded and have results, while a discrete number of trials are ongoing. In 2013, the results of the first-in-man phase I trial involving cancer patients with liver involvement treated with RNA interference (RNAi) molecules encapsulated in LNPs were published. Patients received a combination of two different short interfering RNAs (siRNAs), ALN, and VSP, targeting vascular endothelial growth factor-A (VEGF-A) and kinesin spindle protein encoded by the KIF11 gene (KPS). Similar to other liposomes of the same size (80–100 nm), ALN-VSP distributes primarily to the liver and spleen because of the fenestrated endothelium of these organs. Patients were enrolled in seven dose levels (0.1–1.5 mg/kg) in the dose-escalation phase, followed by an expansion phase. Therapy was administered intravenously every two weeks. Thirty-seven patients were evaluated for tumor response, and four had disease control over six months, with doses ≥ 0.7 mg/kg. A complete response occurred in a patient with endometrial cancer and multiple hepatic and abdominal lymph node metastases, with 50 doses of treatment received over 26 months. Seven patients had a long, stable disease and were treated for an average of 11.3 months. Some of these patients had VEGF-overexpressing primaries, such as renal, endometrial, pancreatic neuroendocrine tumors and angiosarcomas. Two patients with renal cancer experiencing progressive disease during VEGF pathway inhibitors had stable disease for 8–12 months. A patient with a primitive neuro-ectodermal tumor (PNET) and multiple liver metastases had disease stabilization for 18 months. Dynamic contrast-enhanced magnetic resonance imaging (MRI) showed a reduction in blood flow and capillary permeability following ALN-VSP treatment in 13 of the 28 evaluable patients. Moreover, biopsies were performed to look for evidence of RNAi. Among the 12 evaluable patients, all had detectable VEGF siRNA post-dose, and 11 of 12 had detectable KSP siRNA. Two patients had 96–100% viable tumor biopsies, suggesting selective drug delivery to the tumor. Treatment was well tolerated, with the most frequent adverse events being low-grade fatigue/asthenia, nausea/vomiting and fever occurring in 15–24% of patients. Liver function tests were normal in almost all patients, while one patient developed hepatic failure several days after the second dose and subsequently died. Data with ALN-VSP suggest that dual targeting of VEGF and KSP occurs at molecular and clinical levels. These preliminary data provide the basis for further development of ALN-VSP in anti-VEGF responsive malignancies, such as endometrial cancer, PNET, renal cell cancer, and hepatocellular carcinoma (HCC) [9]. Another phase I trial evaluated the activity of polo-like kinase 1 (PLK1) siRNA (TKM-080301) in patients with refractory adrenocortical cancer. The drug was evaluated in a dose-escalation plus expansion cohort and administered an intravenous infusion once a week for three consecutive weeks, repeated every 28 days. Sixteen patients were treated at 0.6 or 0.75 mg/kg/week for up to 18 cycles. Eight patients received at least two cycles and were evaluated for tumor response. Four patients had stable disease, while one patient had a partial response; evaluation of tumor tissue revealed mutations of *TP53*, *NF1,* and *ARID1A*. Moreover, RNA sequencing showed elevated expression of *PLK1* compared to the normal adrenal RNA control. Two patients completed at least six cycles. Treatment was well tolerated, with the most frequent adverse events (AEs) being pyrexia (56%), chills (50%), back pain (31%), infusion reaction (31%), and nausea (25%) [137]. Later, TKM-080301 was evaluated in 43 HCC patients. Patients were enrolled in a 3 + 3 dose escalation plus expansion cohort. The maximum tolerated dose was 0.6 mg/kg. Of the 39 patients receiving a dose of at least 0.3 mg/kg, 18 subjects had stable disease according to RECIST 1.1 criteria, while eight subjects had a partial response according to Choi criteria. The median progression-free survival (PFS) was 2.04 months, while the median survival (mOS) for the whole population was 7.5 months. Treatment was well tolerated but did not improve OS compared to historical control for HCC [138].

The *MYC* oncogene has also been a target of the siRNA therapeutic approach. Dicerna Pharmaceutical developed DCR-MYC, a lipid particle with double-stranded RNA targeting the *MYC* oncogene and suppressing cancer progression [139]. DCR-MYC was administered in a phase I dose-escalation study as a two-hour intravenous infusion on days 1 and 8 of a 21-day cycle. Nineteen patients with advanced solid tumors, multiple myeloma, and lymphoma were enrolled. Considering the role played by *MYC* in cancer metabolism, FDG-PET was performed after one cycle to evaluate metabolic responses. At the same time, tumor shrinkage measured with RECIST criteria was assessed every two cycles. A patient with PNET had dose-limiting toxicity after one cycle at 0.1 mg/kg with a transient increase of aspartate aminotransferase and fatigue. However, this patient obtained a complete metabolic response sustained for more than eight months without further treatment. The most common AEs were fatigue, nausea, and infusion reactions [140]. DCR-MYC was also evaluated in a phase I study of patients with HCC (NCT02314052). However, the producing company has stopped clinical studies with DCR-MYC due to unsatisfactory clinical results [139].

Some trials using RNA-loaded NPs are ongoing. A nanoliposomal targeted therapeutic against EphA2 has been developed to reduce cancer cell proliferation and slow tumor growth. EphA2 is a tyrosine-kinase receptor with overexpression in breast, lung, prostate, ovarian, pancreatic, and endometrial cancer. A phase I study evaluating the role of EphA2 siRNA in patients with advanced solid tumors is currently active but not recruiting (NCT01591356). The companies ModernaTX and AstraZeneca have developed LPNs loaded with mRNA-2752. This mRNA encodes for OX40L, a T-cell co-stimulator. Patients with solid tumors have received mRNA-2752 or a combination with durvalumab. Responses in the preclinical setting using mRNA-2752 in monotherapy or in combination with durvalumab led to a phase I study activation in participants with relapsed/refractory solid tumor malignancies or lymphoma recruiting (NCT03739931).

Another LNP encapsulating mRNA, mRNA-2416, is being evaluated in treating patients with solid malignancies in a phase I-II study alone and in combination with fixed doses of durvalumab (NCT03323398). MRNA-2416 targets OX40, which strongly activates T-cell responses against tumoral cells. Lipo-MERIT is a cancer vaccine of four mRNAs encoding for NY-ESO-1, MAGE-A3, tyrosinase, and PTE. Lipo-MERIT has a splenic tropism, is taken up by dendritic cells and macrophages, and activates NK, B, CD4+ and CD8+ T-cells. Lipo-MERIT is now under evaluation in a phase I study of patients with advanced melanoma (NCT02410733). The mRNA-4157 LNP cancer vaccine is being tested in a phase I study (KEYNOTE-603) alone in patients with resected solid tumors and in combination with pembrolizumab in patients with unresectable solid tumors. After administration, this LNP is uptaken by antigen-presenting cells and induces cytotoxic T-lymphocyte and memory T-cell-dependent immune activation (NCT03313778). In contrast, KEYNOTE-942 is testing mRNA-4157 LNP together with pembrolizumab in patients with cutaneous melanoma completely resected and with a high risk of recurrence (NCT03897881). The OLIVIA trial is a phase-I study currently evaluating a combination of three ovarian cancer tumor-associated antigen mRNAs encapsulated in a liposome. Patients with ovarian cancer are vaccinated during neoadjuvant and adjuvant chemotherapy (e.g., carboplatin-paclitaxel) to increase the T-cell-mediated systemic immune response against cancer cells (NCT04163094). Another phase I study (TNBC-MERIT), currently active but not recruiting, is evaluating the treatment outcome with a combination of immunotherapy with patient-specific RNA-liposome complexes tailored to the antigen-expression profile of any given patient’s tumor (WAREHOUSE immunotherapy–IVAC_W_bre1_uID) and de novo synthesized RNAs targeting up to 20 individual tumor mutations (IVAC^®^ MUTANOME immunotherapy–IVAC_M_uID). The scientific rationale for combining the two IVAC^®^ approaches assumes that immunotherapies that acknowledge tumor heterogeneity on a single-patient level and target the whole range of antigens selectively expressed on tumors bear the highest potential to constitute an effective treatment (NCT02316457). Table 3 lists the major features of the ongoing studies in this field.

## 5. Conclusions

For clinical translatability of promising nano-enabled technologies, well-defined initiatives have already been initiated aiming to improve patient-specific therapeutic outcomes. We have witnessed encouraging results in laboratory settings related to delivery strategies for RNA molecules. However, delivery of tumor suppressor non-coding RNAs is still in its infancy. There is a need to identify highly promising miRNAs, lncRNAs, and circular RNAs for delivery in animal models for disease prevention and treatment. Through continuous and tireless investigation of nanoparticle delivery technologies in laboratory settings, researchers have unique opportunities to conceptually interpret and analyze outcomes to add to an ever-expanding library of known design-function relationship trends in the rapidly evolving field of nanomedicine. However, it is worthwhile that the trends observed at the benchtop must be contextualized before making any attempt to generalize experimental evidence broadly, as minor differences in nanoparticle compositions, animal models, and disease pathology may modify the functions of nanoparticles and must be considered during the clinical translation of nanoparticle technology.

## Figures and Tables

**Figure 1 cancers-14-02677-f001:**
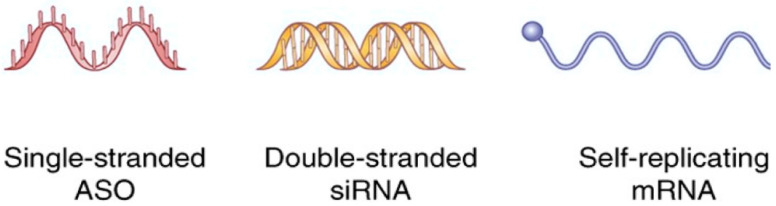
Main RNA used for cancer therapy.

**Figure 2 cancers-14-02677-f002:**
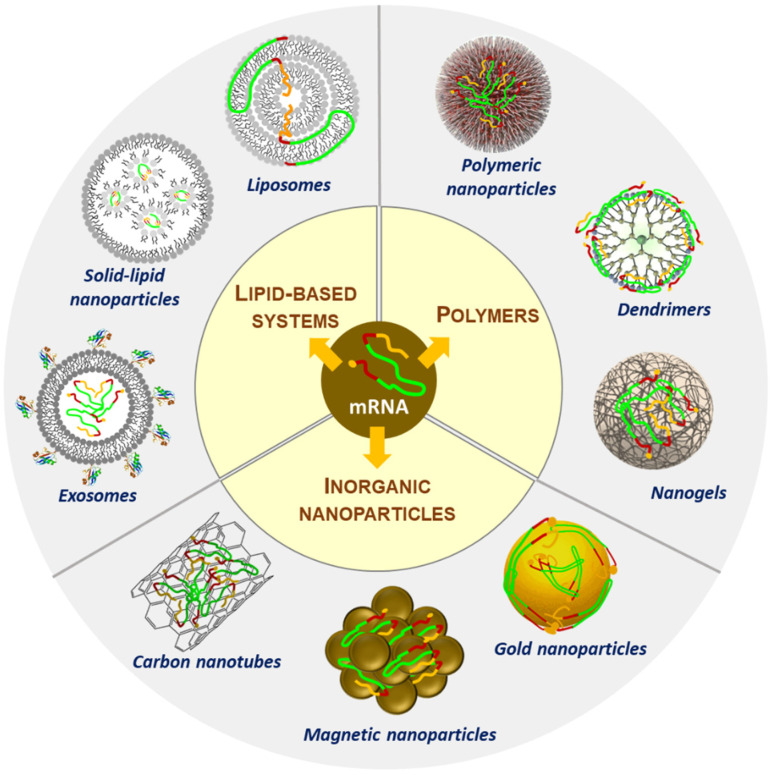
Schematic representation of nanoparticles used in RNA delivery.

**Figure 3 cancers-14-02677-f003:**
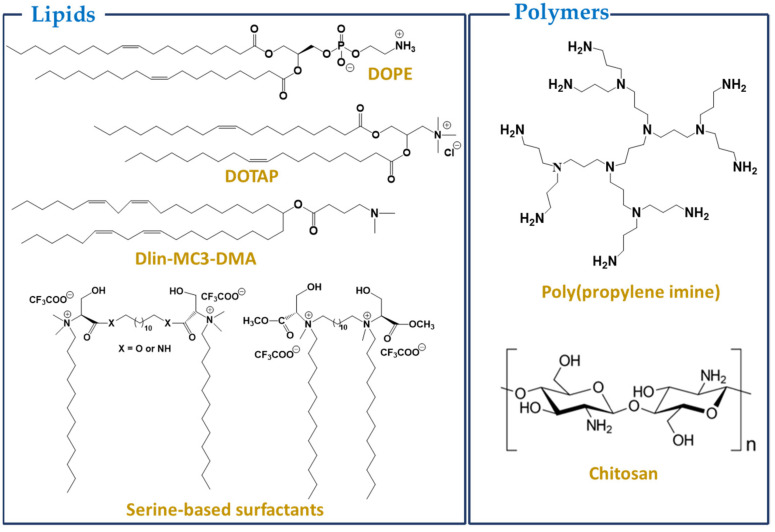
Structures of some common lipids/polymers used for preparing nanoparticles.

**Table 2 cancers-14-02677-t002:** Advantages and disadvantages of nanosystems used for RNA delivery.

Nanosystems		Advantages	Disadvantages	References
Lipid-based nanosystems	Liposomes	- high biodegradability - cationic systems present high compaction efficiency	- cost-effectiveness - off-target effects - cytotoxicity of cationic systems	[43]
	SLN	- easy large-scale production - higher loading capacity than liposomes - high cargo bioavailability - controlled cargo release	- high toxicity when derived from cationic lipids - crystallization upon long-term storage - stability issues	[43,65,66]
	Exosomes	- low immunogenicity - high biocompatibility - high protection of nucleic acids - cell targeting capacity	- heterogeneity of natural matrix - limited large-scale production - limited transfection efficiency	[72]
Polymeric systems		- ease of synthesis and surface modification - versatility of structural conformations - biodegradability (in some cases through derivatization)	- non-degradable polymers tend to accumulate in tissues inducing cytotoxicity - in vivo metabolism and elimination routes still unknown	[134]
Inorganic nanoparticles	Metallic (Au, iron)	- variability in size, structure and geometry - ease of functionalization	- limited information concerning biocompatibility and cytotoxicity	[113,135,136]
	Carbon nanotubes	- possibilities of surface modification - low cytotoxicity and good biocompatibility of modified systems - high loading capacity	- limited in vivo studies developed	[115,116]
	Silica	- large surface area and thus an enhanced loading capacity - low toxicity	- not able to induce endosomal escape (must be chemically modified) - limited information about biocompatibility and biodistribution available	[124,125]

**Table 3 cancers-14-02677-t003:** Ongoing studies with RNA-loaded nanoparticles.

Study Name	Phase/Status	Drug	Target	Indication
NCT01591356	I/Active, not recruiting	EphA2 siRNA	EphA2	advanced/recurrent solid tumors
NCT03739931	I/Active and recruiting	mRNA-2752 LNP	OX40L T cell	relapsed/refractory solid tumor or lymphoma
NCT03323398	I-II/Active, not recruiting	mRNA-2416 LNP alone or + durvalumab	OX40L T cell	relapsed/refractory solid tumor or lymphoma
NCT02410733 Lipo-MERIT	I/Active, not recruiting	mRNA RBL001.1, RBL002.2, RBL003.1, RBL004 LIP	NY-ESO-1, MAGE-A3, tyrosinase and TPTE	advanced melanoma
NCT03313778 KEYNOTE-603	I/Active and recruiting	mRNA-4157 LNP alone or +pembrolizumab	20 TAA	unresectable solid tumor
NCT03897881 KEYNOTE-942	I/Active and recruiting	mRNA-4157 LNP + pembrolizumab	20 TAA	resected and high-risk melanoma
NCT04163094 OLIVIA	I/Active and recruiting	W_ova1	3 TAA	resectable ovarian cancer (neoadjuvant and adjuvant)

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
