# Peer review of "Nanomedicine for the Delivery of RNA in Cancer"

_cancers, 2022, doi:10.3390/cancers14112677_

Round 1

Reviewer 1 Report

  1. In section 2, the difference between these classes of RNA therapies needs to be heightened including their advantages and disadvantages. The contents of table 1 also need to be mentioned in the text.
  2. Peptide and Protein are important for the nanomedicine for the RNA delivery. Should be divided into a section for discussion, and also should include the clinical trial studies of the peptide and protein nanocarrier for RNA delivery in cancer.
  3. In line 114, “nanomedicine” suggests “nanoparticles” instead.
  4. One more figure should be added to reflect the advantages and disadvantages of the nanocarriers, as well as the comparison among them.
  5. In line 136-138, “Although effective … scale-up manufacturing”, authors listed some disadvantages of viral delivery, related references need to be added. Similarly, references to the advantages of non-viral delivery systems need to be added.
  6. In line 145, the authors mentioned the nanoparticle platforms including protein and nucleic acid-based nanoparticles. However, thesetwo nanoparticles were not described below and were not shown in Figure 1. Authors need to limit the scope of the nanoparticles.
  7. In line 226, the authors mentioned PLA, but PLA was not described in the following text. The authors should think about how to deal with this problem. In addition, the polymers described below should be in the same order as the polymers mentioned in line 226.
  8. In section 4, only RNA-Loaded LNPs were described because other RNA-Loaded NPs have not yet entered clinical trials. If so, the authors need to mention it at the beginning of this section.
  9. Any RNA-based selection of nanocarrier for the delivery. What about the siRNA and mRNA with totally different sizes and molecular weights. It is better to include a discussion of the selection of nanocarriers based on the properties of types of RNA.

Author Response

Referee 1

Specific comments:

Comment 1:

In section 2, the difference between these classes of RNA therapies needs to be heightened including their advantages and disadvantages. The contents of table 1 also need to be mentioned in the text.

Response to referee: Thanks for the precious comment. We added a sentence regarding the main features of ASOs. For the other classes included in the review, the main features are listed in section 2. Indeed, siRNAs are double-stranded RNA molecules with 100% target complementarity and high specificity. shRNA are double-stranded and can be incorporated into plasmid vectors, with prolonged knockdown of the target mRNA. On the other hand, miRNAs come from a single-stranded RNA and due to the imperfect base pairing, miRNAs action can affect hundreds of less specific genes. Moreover, we mentioned table 1 in the text.

Comment 2:

Peptide and Protein are important for the nanomedicine for the RNA delivery. Should be divided into a section for discussion, and also should include the clinical trial studies of the peptide and protein nanocarrier for RNA delivery in cancer.

Response to referee: Evidence on peptide and protein nanocarrier for RNA is mainly preclinical (cell lines) while clinical trials in this field are not available. Indeed, although cell penetrating peptides have been shown to be very effective at delivering RNAs into potentially any cell/tissue type, their main disadvantage is that they could not selectively deliver the RNA cargo. This could be detrimental in terms of therapeutics, especially if the peptide-RNA complex cannot distinguish between normal and altered cells/tissues, as this could potentially lead to unwanted cellular toxicities (Cummings, Transl Res 2019)

Comment 3:

In line 114, “nanomedicine” suggests “nanoparticles” instead.

Response to referee: The authors acknowledge the referee for the suggestion and this suggestion was accepted in the new version of the manuscript.

Comment. 4:

One more figure should be added to reflect the advantages and disadvantages of the nanocarriers, as well as the comparison among them.

Response to referee: Two new figures were added in the new version of the manuscript.

Comment 5:

In line 136-138, “Although effective … scale-up manufacturing”, authors listed some disadvantages of viral delivery, related references need to be added. Similarly, references to the advantages of non-viral delivery systems need to be added.

Response to referee: The reviewer is right, missing references have been added.

Comment 6

In line 145, the authors mentioned the nanoparticle platforms including protein and nucleic acid-based nanoparticles. However, these two nanoparticles were not described below and were not shown in Figure 1. Authors need to limit the scope of the nanoparticles.

Response to referee: Thank you for the comment, indeed text and figure must be related. We altered the nanoparticle platforms mentioned in the text so that they match those present in figure 2 (a new Figure 1 was inserted in the manuscript).

Comment 7

In line 226, the authors mentioned PLA, but PLA was not described in the following text. The authors should think about how to deal with this problem. In addition, the polymers described below should be in the same order as the polymers mentioned in line 226.

Response to referee: PLA was removed (we do not aim at an exhaustive description of all the existing nanoplatforms) and the order of the polymers was changed in the text so that it confers with the order they are described below.

Comment 8

In section 4, only RNA-Loaded LNPs were described because other RNA-Loaded NPs have not yet entered clinical trials. If so, the authors need to mention it at the beginning of this section.

Response to referee: The authors are grateful for the valuable suggestion and this information was inserted in the new version of the manuscript.

Comment 9

Any RNA-based selection of nanocarrier for the delivery. What about the siRNA and mRNA with totally different sizes and molecular weights. It is better to include a discussion of the selection of nanocarriers based on the properties of types of RNA.

Response to referee: Thanks for the precious comment. We added a short discussion on different molecular weights between siRNAs and mRNAs at the end of section 2.

Reviewer 2 Report

It was a review article about the application of nanomedicine for the delivery of different types of RNA for cancer therapy. here are some comments on this study that should be considered before publication:

1. Please improve the quality of the introduction section.

2. There are some grammatical mistakes in the text that should be corrected.

3. “Among ASO, some of them act as miRNAs inhibitors. These oligonucleotides bind to the active chains of endogenous miRNAs with gene-silencing effects. Therefore, they enhance gene expression.” is this true?

4. “disrupt the expression” the expression of what?

5. “RNA therapeutics may act through several mechanisms. They can inhibit the proliferation and induce apoptosis of tumour cells, prevent the metastasization process, disrupt the expression, inhibit angiogenesis, reconstruct the tumour environment, reprogram and decrease drug resistance of tumour cells [2].” please transfer these sentences to the first parts of section 2.

6. Please mention references in a separate column in table 1.

7. Please add references related to section 3. Most of the parts of this section had no reference.

8. “The translation efficiency and stability of exogenous RNA can be enhanced by several methods such as UTR (untranslated regions) manipulation, codon optimization, chemical modification of poly(A)tail of RNA; furthermore, its immunogenicity can be reduced through high-performance liquid chromatography purification and chemical manipulation.” this paragraph could be divided to two different sentences.

9. “Several cationic LNP have been successfully used as RNA carriers in targeted cancer therapy, leading to higher accumulation and increased protein expression, which resulted in suppressed/blocked tumour growth. [12, 26]” please add more references here. The same for the following sentences

- “Several MC3-based LNP have then been tested for mRNA therapeutics.”

- “The biodegradability may be conferred by the presence of an ester bond on the hydrophobic tail or on the linker, which accelerates liver clearance. In addition, the release of the cargo may be triggered through cleavage of the labile ester bond (pH, nucleases) and consequent modification of the aggregate structure.”

- “Polymeric based non-viral vectors represent another class of nanoscale platforms for RNA delivery. Specifically, cationic polymers can bind to nucleic acids to form polyplexes.  Polymers can efficiently protect RNA from nucleases, promote cellular uptake and endosomal escape, leading to higher RNA delivery efficiency. Representative polymers of this class include polylactic acid (PLA), polyethyleneimine (PEI), chitosan, dendrimers and polymethacrylates. Chitosans are naturally derived cationic polysaccharides, differing in the degree of N-acetylation and molecular weight (50-2000 kDa). They are readily available, biodegradable, easy to modify and possess unique biological properties associated with their polycationic nature. Chitosan itself is only poorly water-soluble and exhibits low transfection efficacy. However, it can be derivatized to increase nucleic acid delivery efficiency by the chitosan vectors. Strategies for derivatization include structural modifications – like i) copolymerization: polyethylene glycol, PEG, and polyethyleneimine, PEI, are commonly used, although other chitosan graft copolymers are being studied as nucleic acid carriers; and ii) functional group modification: N-alkylation and quaternization enhance colloidal stability and transfection efficacy of the nanoparticles – and ligand conjugation – peptides, proteins and non-proteinaceous ligands, like carbohydrates, folic acid and hyaluronic acid are commonly used for chitosan vector conjugation. Although non-proteinaceous ligands are usually less immunogenic and produce more stable vectors, proteinaceous ligands offer a vast diversity of choices with favourable functionalities to vector conjugation for nucleic acid delivery.”

- “Dendrimers are polymeric materials with a highly branched 3D structure. They consist of a central core, many layers of repeating units and multiple functional groups on the surface. Due to their unique structure, they possess various interesting physical properties, such as good water-solubility, nanoscale uniform size, symmetrical shapes, internal cavities, good biocompatibility, stability and high drug-loading capacity. Their toxicity is generally low but depends on the number of terminal amino groups and positive charge density. Dendrimer-based organic and inorganic nanoparticles have been widely studied and exhibited a high potential in cell targeting and drug delivery.”

- “Inorganic nanomaterials (derived from gold, carbon, silica, etc.) are promising carrier platforms for RNA delivery, due to their unique physicochemical properties, which endow them exciting attributes, such as long-term stability, high loading capacity and optical responsiveness. These inorganic nanoparticles are usually easier to synthesize and scale-up than the organic-based one and have been subject of much research concerning their potential as nanocarriers for nucleic acid delivery.”

- …

You can use the following references:

- https://www.mdpi.com/1381542

- https://www.sciencedirect.com/science/article/pii/S014486172100196X

- https://pubs.acs.org/doi/abs/10.1021/acscombsci.0c00099

- https://www.mdpi.com/793562

10. Please explain more about the samples mentioned in section 3. Explain their effect on cancer treatment. Moreover, please add some good figures of the used references for a better explanation.

11. Please add other clinical samples in which other types of nanomaterials are used for the delivery of RNAs.

12. Please improve the quality of the conclusion part.

Author Response

Reviewer 2:

It was a review article about the application of nanomedicine for the delivery of different types of RNA for cancer therapy. here are some comments on this study that should be considered before publication:

  1. Please improve the quality of the introduction section.

Response to referee: The authors are grateful for the suggestion and the introduction section was improved.

  1. There are some grammatical mistakes in the text that should be corrected.

Response to referee: The manuscript was carefully revised to correct typos and grammatical mistakes.

  1. “Among ASO, some of them act as miRNAs inhibitors. These oligonucleotides bind to the active chains of endogenous miRNAs with gene-silencing effects. Therefore, they enhance gene expression.” is this true?

Response to referee: Thanks for the useful comment. miRNA inhibitors are chemically modified ASOs that weaken the silencing effect of endogenous miRNAs by specifically binding to the active chains of them and increase protein expression. We added another reference [Y. Weng, H. Xiao, J. Zhang, X.J. Liang, Y. Huang, RNAi therapeutic and its innovative biotechnological evolution, Biotechnol. Adv. 37 (5) (2019) 801–825] to the existing one [Liang, X., et al., RNA-based pharmacotherapy for tumors: From bench to clinic and back. Biomed Pharmacother, 2020. 125: p. 109997]

  1. “disrupt the expression” the expression of what?

Response to referee: This sentence was corrected to be clearer and to avoid any subjective interpretation.

  1. “RNA therapeutics may act through several mechanisms. They can inhibit the proliferation and induce apoptosis of tumour cells, prevent the metastasization process, disrupt the expression, inhibit angiogenesis, reconstruct the tumour environment, reprogram and decrease drug resistance of tumour cells [2].” please transfer these sentences to the first parts of section 2.

Response to referee: The authors agree with referee’s comment and this sentence was transferred in the revised version of the manuscript.

  1. Please mention references in a separate column in table 1.

The authors thank the suggestion provided and the references were added in a separate column.

  1. Please add references related to section 3. Most of the parts of this section had no reference.

Response to referee: References were added to section 3 and respective subsections.

  1. “The translation efficiency and stability of exogenous RNA can be enhanced by several methods such as UTR (untranslated regions) manipulation, codon optimization, chemical modification of poly(A)tail of RNA; furthermore, its immunogenicity can be reduced through high-performance liquid chromatography purification and chemical manipulation.” this paragraph could be divided to two different sentences.

Response to referee: For clarity, the paragraph was divided in two different sentences.

  1. “Several cationic LNP have been successfully used as RNA carriers in targeted cancer therapy, leading to higher accumulation and increased protein expression, which resulted in suppressed/blocked tumour growth. [12, 26]” please add more references here. The same for the following sentences

- “Several MC3-based LNP have then been tested for mRNA therapeutics.”

- “The biodegradability may be conferred by the presence of an ester bond on the hydrophobic tail or on the linker, which accelerates liver clearance. In addition, the release of the cargo may be triggered through cleavage of the labile ester bond (pH, nucleases) and consequent modification of the aggregate structure.”

- “Polymeric based non-viral vectors represent another class of nanoscale platforms for RNA delivery. Specifically, cationic polymers can bind to nucleic acids to form polyplexes.  Polymers can efficiently protect RNA from nucleases, promote cellular uptake and endosomal escape, leading to higher RNA delivery efficiency. Representative polymers of this class include polylactic acid (PLA), polyethyleneimine (PEI), chitosan, dendrimers and polymethacrylates. Chitosans are naturally derived cationic polysaccharides, differing in the degree of N-acetylation and molecular weight (50-2000 kDa). They are readily available, biodegradable, easy to modify and possess unique biological properties associated with their polycationic nature. Chitosan itself is only poorly water-soluble and exhibits low transfection efficacy. However, it can be derivatized to increase nucleic acid delivery efficiency by the chitosan vectors. Strategies for derivatization include structural modifications – like i) copolymerization: polyethylene glycol, PEG, and polyethyleneimine, PEI, are commonly used, although other chitosan graft copolymers are being studied as nucleic acid carriers; and ii) functional group modification: N-alkylation and quaternization enhance colloidal stability and transfection efficacy of the nanoparticles – and ligand conjugation – peptides, proteins and non-proteinaceous ligands, like carbohydrates, folic acid and hyaluronic acid are commonly used for chitosan vector conjugation. Although non-proteinaceous ligands are usually less immunogenic and produce more stable vectors, proteinaceous ligands offer a vast diversity of choices with favourable functionalities to vector conjugation for nucleic acid delivery.”

- “Dendrimers are polymeric materials with a highly branched 3D structure. They consist of a central core, many layers of repeating units and multiple functional groups on the surface. Due to their unique structure, they possess various interesting physical properties, such as good water-solubility, nanoscale uniform size, symmetrical shapes, internal cavities, good biocompatibility, stability and high drug-loading capacity. Their toxicity is generally low but depends on the number of terminal amino groups and positive charge density. Dendrimer-based organic and inorganic nanoparticles have been widely studied and exhibited a high potential in cell targeting and drug delivery.”

- “Inorganic nanomaterials (derived from gold, carbon, silica, etc.) are promising carrier platforms for RNA delivery, due to their unique physicochemical properties, which endow them exciting attributes, such as long-term stability, high loading capacity and optical responsiveness. These inorganic nanoparticles are usually easier to synthesize and scale-up than the organic-based one and have been subject of much research concerning their potential as nanocarriers for nucleic acid delivery.”

- …

You can use the following references:

- https://www.mdpi.com/1381542

- https://www.sciencedirect.com/science/article/pii/S014486172100196X

- https://pubs.acs.org/doi/abs/10.1021/acscombsci.0c00099

- https://www.mdpi.com/793562

Response to referee: References were added to section 3 and respective subsections.

  1. Please explain more about the samples mentioned in section 3. Explain their effect on cancer treatment. Moreover, please add some good figures of the used references for a better explanation.

Response to referee: The authors are grateful for the referee’s suggestion, which it is very pertinent. However, this manuscript is focus on cancer prevention and treatment using RNA therapeutics loaded in lipid nanoparticles, which seems to be the most promising strategy for the delivery of genetic material. The use of nanomedicine and particularly lipid nanoparticles is a suitable way to protect and to deliver RNA. The effect of the nanoparticles by themselves in cancer is not the focus since the mechanism in cancer is related with the RNA delivered and the idea is to protect and deliver the different types of RNA without any toxicity, which was already explained in detail in the other sections.

  1. Please add other clinical samples in which other types of nanomaterials are used for the delivery of RNAs.

Response to referee: No other types of nanomaterials are currently used in the clinical practice and even in clinical trials and this information was inserted in the revised version of the manuscript.

  1. Please improve the quality of the conclusion part.

Response to referee:The authors are grateful for the suggestion and the conclusion section was improved.

Round 2

Reviewer 1 Report

Thank you for your revisions, and my concerns are all addressed.

One more thing is regarding Q2. There might be peptide molecular for RNA therapy that has been applied in the clinical trial. You might double-check and confirm.

Reviewer 2 Report

-